# Medical Fuzzy-Expert System for Assessment of the Degree of Anatomical Lesion of Coronary Arteries

**DOI:** 10.3390/ijerph20020979

**Published:** 2023-01-05

**Authors:** Waldemar Wójcik, Iryna Mezhiievska, Sergii V. Pavlov, Tomasz Lewandowski, Oleh V. Vlasenko, Valentyn Maslovskyi, Oleksandr Volosovych, Iryna Kobylianska, Olha Moskovchuk, Vasyl Ovcharuk, Anna Lewandowska

**Affiliations:** 1Faculty of Electrical Engineering and Computer Science, Lublin University of Technology, Nadbystrzycka 38d, 20-618 Lublin, Poland; 2Department of Internal Medicine No. 3, National Pirogov Memorial Medical University, Pirogov Str. 56, 21018 Vinnytsya, Ukraine; 3Laboratory of Biomedical Optics, Faculty for Infocommunications, Radioelectronics and Nanosystems, Vinnytsia National Technical University, Khmelnytske Shose 95, 21021 Vinnytsia, Ukraine; 4Institute of Technical Engineering, State School of Technology and Economics in Jaroslaw, 37-500 Jaroslaw, Poland; 5Laboratory of Experimental Neurophysiology, National Pirogov Memorial Medical University, 21018 Vinnytsia, Ukraine; 6Department of Biomedical Engineering and Optic-Electronic Systems, Vinnytsia National Technical University, Khmelnytske Shose 95, 21021 Vinnytsia, Ukraine; 7Department of Life Safety and Safety Pedagogy, Vinnytsia National Technical University, Khmelnytske Shose 95, 21021 Vinnytsia, Ukraine; 8Department of Pedagogy, Vinnytsia Mykhailo Kotsiubynskyi State Pedagogical University, Ostrozhsky Str. 32, 21000 Vinnytsia, Ukraine; 9Department of Physical Education, Vinnytsia National Technical University, Khmelnytske Shose 95, 21021 Vinnytsia, Ukraine; 10Institute of Healthcare, State University of Technology and Economics in Jaroslaw, Czarniecki Street 16, 37-500 Jaroslaw, Poland

**Keywords:** medical expert systems, fuzzy logic, coronary artery disease, coronary arteries, problems of cardiology, cardiovascular diseases, myocardial infarction, patient safety

## Abstract

Background: Today, cardiovascular diseases cause 47% of all deaths among the European population, which is 4 million cases every year. In Ukraine, CAD accounts for 65% of the mortality rate from circulatory system diseases of the able-bodied population and is the main cause of disability. The aim of this study is to develop a medical expert system based on fuzzy sets for assessing the degree of coronary artery lesions in patients with coronary artery disease. Methods: The method of using fuzzy sets for the implementation of an information expert system for solving the problems of medical diagnostics, in particular, when assessing the degree of anatomical lesion of the coronary arteries in patients with various forms of coronary artery disease, has been developed. Results: The paper analyses the main areas of application of mathematical methods in medical diagnostics, and formulates the principles of diagnostics, based on fuzzy logic. The developed models and algorithms of medical diagnostics are based on the ideas and principles of artificial intelligence and knowledge engineering, the theory of experiment planning, the theory of fuzzy sets and linguistic variables. The expert system is tested on real data. Through research and comparison of the results of experts and the created medical expert system, the reliability of supporting the correct decision making of the medical expert system based on fuzzy sets for assessing the degree of anatomical lesion of the coronary arteries in patients with various forms of coronary artery disease with the assessment of experts was 95%, which shows the high efficiency of decision making. Conclusions: The practical value of the work lies in the possibility of using the automated expert system for the solution of the problems of medical diagnosis based on fuzzy logic for assessing the degree of anatomical lesion of the coronary arteries in patients with various forms of coronary artery disease. The proposed concept must be further validated for inter-rater consistency and reliability. Thus, it is promising to create expert medical systems based on fuzzy sets for assessing the degree of disease pathology.

## 1. Introduction

Coronary artery disease (CAD) remains one of the leading causes of temporary and permanent disability and mortality of the population in economically developed countries and is one of the most urgent problems of cardiology [1]. 

Today, cardiovascular diseases cause 47% of all deaths among the European population, which is 4 million cases every year [2]. In Ukraine, CAD accounts for 65% of the mortality rate from diseases of the circulatory system of the able-bodied population and is the main cause of disability [3,4].

Despite the fact that in Europe, the mortality rate associated with coronary artery disease has decreased over the past decades, this pathology remains one of the leading causes of death. Relative indices of STEMI rates are decreasing, while of NSTEMI are correspondingly increasing. Despite the decrease in mortality associated with STEMI, accompanied by the expansion of the practice of reperfusion therapy, mortality remains significant. Hospital mortality in these patients, according to European registers, ranges from 4% to 12% [5,6,7]. 

Myocardial infarction is the most frequent manifestation of coronary artery disease and one of the main causes of disability and mortality in the working population. The mortality rate in myocardial infarction is 18.5–40% and a significant number of patients die at the onset of the attack, in most cases before hospitalisation. Over the past 10 years, the incidence of NSTEMI (non-ST segment elevation myocardial infarction) has increased significantly. An important point in the introduction of patients with NSTEMI is the development of stratification and prediction of the course, using various clinical and instrumental parameters. 

Systematised data on the nature of coronary artery lesions in patients with NSTEMI demonstrate that 10–20% of patients have intact coronary arteries, in 30–35% of cases there is a lesion of one, in 25–30%—2 arteries, and in 5–10%—lesions trunk of the left coronary artery of various degrees [8,9,10,11]. Several studies demonstrate less significant anatomical changes in coronary arteries in women, compared to men in all age groups [12,13,14,15].

In Ukraine, the frequency of registration of myocardial infarction (MI) is one of the highest in the European population and is about 50,000 cases annually, which gives reason to consider this pathology as one of the priority medical and social problems of our society. According to the fourth universal definition, the term “myocardial infarction” is used in the presence of proven myocardial necrosis due to acute prolonged myocardial ischemia [16]. Namely, the activation of the structural and geometric rearrangement of the heart chambers is associated with necrosis and subsequent fibrosis of the myocardium, which causes a number of problems in the post-infarction period. In long-term MI, the survival prognosis of patients is mainly determined by the development and progression of heart failure (HF), the basis of which is left ventricular dysfunction (LV) [17,18].

In the estimation of scientists, by 2030, 44% of the population will have some type of cardiovascular disease (CVD) [18]. Patients with CVD experience numerous subjective symptoms, including fatigue, shortness of breath, or chest pain, which affect their physical, emotional, and social status with significant impairment of quality of life [19,20,21].

Therefore, an important point in the introduction of patients with NSTEMI is the development of stratification and forecast of the course of the disease, which will provide an opportunity to carry out preventive measures and avoid destabilisation of the course of the disease and the development of various complications.


**Scientific setting and formulation of the aim of the work.**


In order to increase the effectiveness of the assessment of the course of the disease and predict the nature of coronary artery lesions, the authors propose the use of the mathematical theory of fuzzy sets. Various computer systems based on this theory, in turn, significantly expand the scope of application of fuzzy logic, in particular, for the analysis of biomedical data of any complexity. Therefore, these computer fuzzy technologies for biomedicine are of great interest among scientists today in the direction of the development of the method of fuzzy intellectual data analysis as one of the most effective areas of application of the theory of fuzzy sets.

The key advantages of fuzzy logic in comparison with other technologies of intelligent analysis are: firstly, with the same volumes of input and output information, the central decision-making unit becomes more compact and easier for human perception, secondly, the solution of a complex and cumbersome calculation task precise actions are replaced by much simpler and more flexible ones.

An important element of the practical implementation of the theory of fuzzy sets is the possibility of forming fuzzy queries into databases. Mechanisms of fuzzy queries (fuzzy queries, flexible queries) to relational databases were first proposed in 1984 and later developed in the works of D. Dubois and G. Prada.

The basic works on which the principles of the implementation of expert fuzzy systems in assessing the course of the disease and predicting the nature of coronary artery damage are based are the scientific work of Bellman and Zade (the article “Decision-Making in Fuzzy Environment”), which served as the starting point for most works on the fuzzy theory of acceptance solutions. That paper examines the decision-making process under conditions of uncertainty when goals and constraints are set by a fuzzy set [22,23,24,25,26,27,28,29,30,31,32,33].

The authors also rely on the research of Professor Rotshtein A. (Design and Tuning of Fussy IF–THEN Vuly for Medical Didicol Diagnosis. In Fussy and Neuro-Fussy Systems in Medicine), which present the basic models necessary for fuzzy logical inference and rough tuning of fuzzy rules by the method of pairwise comparison as well as fine-tuning as an optimization problem, development of mathematical models for evaluating the quality of fuzzy inference, and analysis of computer experiments for fuzzy models with continuous and discrete consequences, respectively. In this paper, an approach to the construction of a fuzzy expert system for the differential diagnosis of coronary heart disease is proposed. From a formal point of view, the task of building a fuzzy model of medical diagnostics can be considered as the task of identifying a nonlinear object with several inputs and one output [34].

***The aim of this work*** is the development of a medical expert system based on fuzzy sets for assessing the degree of coronary artery lesions in patients with coronary artery disease.

## 2. Methods

### 2.1. The Concept of Predicting the Coronary Arteries Lesion Character Using Clinical and Various Non-Invasive Markers

The following types of myocardial blood supply are distinguished:Left coronary type, in which the greater part of the heart is supplied with blood by branches of the left coronary artery (arteria coronaria sinistra);Right coronary type, in which the greater part of the heart is supplied with blood by branches of the right coronary artery (arteria coronaria dextra);The average type, in which the coronary arteries evenly supply blood to the heart;The intermediate type; it can be:Centre-right;Middle-left.

In the case of the right type of blood supply to the heart, 75% of the blood feeding the myocardium passes through the trunk of the left coronary artery (TrLCA), with the left—almost 100%. In this regard, patients with a hemodynamically significant lesion of the left ventricle have a high risk of death and left ventricular dysfunction and arrhythmias. Anatomically, the TrLCA is divided into the ostium, middle and distal parts. The average diameter of the LCA, according to angiography, is 4.5 ± 0.5 mm for men and 3.9 ± 0.4 mm for women. However, observations have been described when, according to autopsy data, the diameter of the ventricular septal defect reached 1 cm in persons with a healthy heart [1]. According to the data of coronary angiography (CAG), lesions of the TrLCA occur in 4–8% of patients with coronary artery disease (CAD) [22,23].

Most of the data presented in the literature regarding coronary artery lesions in NSTEMI patients indicate that 10–20% of patients have intact coronary arteries, 30–35% of cases have a lesion of one, 25–30%—a lesion of 2 arteries, and 5–10%—lesions of the TrLCA trunk of various degrees [8,24,25,26]. On the other hand, several studies demonstrate less significant changes in coronary arteries in women, compared to men, in various forms of acute coronary syndrome in all age groups [4,27,28]. Of particular interest are studies that show the concept of predicting the nature of coronary artery lesions using clinical and various non-invasive markers, which makes it possible to stratify risks in patients for invasive treatment using simple and affordable research methods [26].

### 2.2. Characteristic Features of Creating Medical Expert Systems for Processing and Organising Medical Data Based on Fuzzy Sets

The application of the theory of fuzzy sets in the context of the implementation of expert fuzzy systems in assessing the course of the disease and predicting the nature of coronary artery lesion has the following advantages compared to others: the ability to operate with fuzzy input data: for example, values that change continuously over time (dynamic problems), values, which cannot be specified unambiguously (results of statistical data); possibility of vague formalisation of evaluation and comparison criteria: operation with vague terms, “low”, “ lower than average”, “average”, “ Higher than average”, “high”; the possibility of conducting quality assessments of both input data and output results: we have the opportunity to operate not only with data values but also with their degree of reliability and its distribution; the possibility of rapid modelling of complex dynamic systems and their comparative analysis with a given degree of accuracy [35,36].

A powerful tool for the practical implementation of fuzzy calculations is the Fuzzy Logic Toolbox, a package of application programs included in the MATLAB environment. It allows you to create systems of fuzzy logic inference and fuzzy classification within the MATLAB environment, with the possibility of their integration in Simulink. The basic concept of Fuzzy Logic Toolbox is the FIS structure—fuzzy inference system (Fuzzy Inference System).

Thus, it is promising to create expert medical systems based on fuzzy sets for assessing the degree of disease pathology.

Medical expert information processing systems use two approaches to the organisation of medical data:FragmentationComplexing [30,31,37].

In case of fragmentation, the task of data processing is divided into separate parts in order to solve it more efficiently. In the process of complexing for the solution of separate problems, the parameters are united in larger sections. In practice, both approaches are used in medical expert systems (MES), since the data of various studies are closely related. Processing results are used to verify the diagnosis, choose treatment methods, predict conclusions, etc.

In the process of developing medical diagnostic and information systems, the analysis of parameters used by modern medicine is of great importance.

While developing a medical expert system, it is necessary to solve a number of tasks, namely:Selection and determination of the system designation;Selection of the structural scheme of the system;Formation and analysis of the list of nosological forms to be studied, collection of statistically reliable information about the severity of symptoms, as well as about the functional state;Selection of a method of medical bioinformation processing;Construction of an algorithm for solving the problems of evaluating medical bioinformation and forming diagnostic and prognostic conclusions [32,33,34].

The design of the expert system will be of high quality only if the research is conducted by an experienced diagnostician. Such research can be designed by a group of qualified experts in this field of diagnostics.

In the last decade, a group of methods has been rapidly developing where the theory of fuzzy sets [38] has been used for classification and on its base fuzzy rules. The membership of an element x to a fuzzy set M is given by the continuous membership function μM(x) (0 ≤ μM(x) ≤ 1). Linguistic variables L, possible values of which are a set of terms T(L), where each term represents a label of a fuzzy set and is specified by its membership function μTi(x), are used to quantify the rules that are close to natural language sentences.

In this way, the transition from the numerical variable x to the linguistic variable L is performed. Logical operations with fuzzy sets and formalised fuzzy inference rules used to form decision rules are defined. The type of membership function μM(x) is specified by the researcher (S, Π, triangular, trapezoidal, etc.), and its parameters are determined on the training sample according to the criterion of the minimum classification error, for which genetic algorithms are widely used. Since the method is based on expert evaluations, it is a serious alternative to probabilistic methods when the training sample is insufficient or absent [38].

In recent years, two classes of mathematical models for medical diagnostics have gained the greatest development: formal or formalised models and models based on fuzzy logic, which along with the class of conceptual models fully cover the subject area of research related to medicine and rehabilitation.

A logical complement can be considered a list of requirements for medical diagnostic models; the sequence of stages of their construction and the formulation of the basic advantages of using mathematical models for diagnostic purposes.

When forming mathematical models, we relied on the principles of Lotfi Zadeh’s theory of fuzzy sets; he moved away from the discrete concept of “membership“ and introduced a new one—“membership degree”, and the usual “set“ was replaced by “fuzzy sets”.

The following main features of the theory of fuzzy sets were taken into account:A fuzzy model (fuzzy set) on the universal set U is a collection of pairs $\left(\mu_{A}(u),u\right)$, where $\left(\mu_{A}(u)\right)$, is the membership degree of the element to the fuzzy set.A membership degree is a number from the range of [0, 1]. The higher the membership degree, the more the element of the universal set corresponds to the properties of this fuzzy set. So, if the degree of membership is 0, then this element does not correspond to the set, and if it is 1, then we can speak, on the contrary, about full correspondence. These two cases are marginal and in the absence of other options would represent the usual set. The presence of all other options is the key difference of the fuzzy set.A membership function is a function that allows you to calculate the degree of membership of an arbitrary element of a universal set to a fuzzy set. Therefore, the range of values of membership functions should belong to the range [0, 1]. In most cases, the membership function is a monotonic continuous function.A linguistic (fuzzy) variable is a variable whose values can be words or phrases of some natural or artificial language. Fuzzy sets are made up of linguistic variables. When defining a fuzzy set, the number and nature of fuzzy variables are subjective for each individual problem.A set of terms (term set) is a set of all possible values that a linguistic variable can take.A term is any element of a term set. In the theory of fuzzy sets, a term is formalised by a fuzzy set using a membership function. The membership function for each term is individual and often unique. There are different methods of constructing these functions: direct, indirect and relative frequency methods. They are most often based on characteristic points of the membership function or empirical data of an expert in a given subject area.Defuzzification is a procedure of converting a fuzzy set into a clear number. At the moment, more than twenty methods are distinguished, and their results can differ significantly from each other. It should be noted that the best results are obtained by defuzzification using the centre of gravity method [33].

The main features of fuzzy logic, which distinguish it from classical logic, are the maximum approximation to the reflection of reality and a high level of subjectivity, as a result of which significant errors may occur in the results of calculations using fuzzy logic.

The quality of the initial values of these models (model error) directly depends only on the expert who compiled and adjusted the model. To minimise the error, the best option would be to compile the most complete and comprehensive model and further adjust it by means of machine learning on a large training sample.

The quality of the output values of these models (model error) directly depends only on the expert who compiled and adjusted the model. To minimise the error, the best option would be to compile the most complete and comprehensive model and further adjust it by means of machine learning on a large training sample.

The model-building process can be divided into three main stages [39,40]:Determination of input and output parameters of the model.Building a knowledge base.Choosing one of the methods of fuzzy logical conclusion.

The other two stages directly depend on the first stage, and it determines the future functioning of the model. A knowledge base or, as it is called differently, the rule base is a set of fuzzy rules of the type: “if, then”, which determine the relationship between the inputs and outputs of the object under study 

Figure 1 shows the suggested structure of MES, containing the basic blocks, for the implementation of the interface, a computer program was developed to perform such functions as receiving feedback from an expert to form the base of the knowledge base and conducting a dialogue with the user.

Based on the results of the research, the architecture of the medical expert system was developed, which has the function of designing and tuning fuzzy knowledge bases, which represent a set of linguistic transformations of the type: if <inputs>, then <outputs>. In this version, the input changes used for data transformation in the expert system can be presented in a qualitative and quantitative form [41].

The proposed medical expert system (see Figure 2) contains a biomedical information conversion unit, a biomedical data input unit and biomedical information preprocessing unit, an expert knowledge accumulation unit, a storage unit for membership functions, and a research results output unit. At the same time, the user has the opportunity to make corrections, replenish the knowledge base, and tune the membership functions. The processing of biomedical information takes place according to a complex scheme of hierarchical transformation of incoming medical information, which is presented in the form of a neural network.

## 3. Results

### 3.1. Formation of Databases for the Implementation of an Expert Decision-Making System for the Assessment of the Degree of Anatomical Lesion of Coronary Arteries

Based on the comprehensive examination of 165 patients with various forms of coronary artery disease (CAD) with/and without hypertension (AH) aged 35 to 79 years (on average 60.7 ± 0.8, median—61, interquartile range—54 and 69) experts analysed the peculiarities of the anatomical lesion of the coronary arteries in patients with various forms of coronary artery disease (see Table 1). Among the examined patients, 114 (69.1%) were male and 51 (30.9%) were female, respectively. The ratio of men to women was 2.2 to 1.0 (χ^2^ = 48.1; *p* < 0.0001), which indicated a significant predominance of male patients.

As the reason for including patients in the study, the following criteria were considered:

1. stable and acute forms of coronary arteries disease (stable angina pectoris II–III FC, unstable angina pectoris and acute myocardial infarction with and without elevation of the ST segment);

2. acute myocardial infarction of the left ventricle (LV), which occurred for the first time (in the absence of a history of MI);

3. age of patients from 30 to 80 years.

The analysis of the main clinical characteristics of *NSTEMI* patients (see Table 2) showed that 85.5% of the examined had GC lasting from 7 to 25 (on average, 15.5 ± 0.41) years. In 43.0% of patients, instrumentally proven (according to medical documents) stable angina pectoris I–III FC with a history of 1 to 15 (on average 7.0 ± 0.44) years was observed before the acute MI incident. A permanent form of atrial fibrillation was registered in 11.5% of the examinees, and the history of permanence of arrhythmia ranged from 1 to 7 and averaged 4.4 ± 0.39 years. Moreover, 12.5% of *NSTEMI* patients had a history of type II diabetes and 42.0% had such a risk factor as smoking. At the same time, the vast majority (80.0%) of these patients smoked at the time of the MI, and only 20.0% were smokers in the past (the period of quitting the habit did not exceed 2 years). The total smoking history ranged from 14 to 40 and averaged 29.5 ± 0.84 years.

The body mass index (BMI) of the investigated patients varied from 19.3 to 47.6 and was 28.6 ± 0.36 kg/m^2^ on average. Alimentary obesity (BMI > 30 kg/m^2^) was determined in 36.5% of patients. At the same time, obesity of the first degree (BMI—30–35 kg/m^2^) was diagnosed in 25.5%, II (BMI—35–40 kg/m^2^)—in 9.0% and III degree (BMI > 40 kg/m^2^)—only in 2.0% of cases.

In its turn, a comparative analysis of the main characteristics of patients depending on gender (table) showed that among males, compared to females, a significant increase in smoking cases was observed (52.1% vs. 17.2%, *p* < 0.0001) and, accordingly, smoking history (29.6 versus 25.0 years, *p* = 0.002). On the other hand, among females, compared to males, a significant increase in cases of preinfarction angina (55.2% vs. 38.0%, *p* = 0.03) and persistent atrial fibrillation was observed (22.4% vs. 7.0%, *p* = 0.002). The last fact was not confirmed in modern literature, although we believe that it can be explained by the significantly longer span of life of women compared to men. At the same time, there is a close connection between the development of atrial fibrillation with the age of patients.

Thus, it should be stated that among *NSTEMI* males such a risk factor as smoking is more often registered, while among females—preinfarction angina and a permanent form of AF is registered as risk factor.

### 3.2. Implementation of a Medical Expert System Based on Fuzzy Sets for the Assessment of the Degree of Anatomical Lesion of the Coronary Arteries

Table 3 was created with the determination of the minimum and maximum values of factors X1–X4. In our case, X1 (the presence of a/p in the basin of DG or LAD), X2 (the presence of HSS in the basin of DG or LAD LCA), X3 (the presence of HSS in the trunk of the RCA), X4 (absence of HSS CA). The main clinical forms of coronary artery disease were determined (STEMI) µ^I^(x_1X2_x_3_x_4_), UAP—unstable angina pectoris, µ^II^(x_1_x_2_x_3_x_4_), STEMI—myocardial infarction with ST segment elevation µ^III^(x_1_x_2_x_3_x_4_), StAP—stable angina pectoris µ^IV^(x_1_x_2_x_3_x_4_).

Taking into account the ranges of factors X1–X4, the knowledge base of experts was formed, based on the knowledge bases of experts.

Such an approach enables to obtain a clear numerical expression for those criteria, that have descriptive characteristics and, correspondingly, qualitative content, for instance, such characteristics as L—low, LA—lower than average; A—average; HA—higher than average, H—high [29,31] (see Table 4).

Corresponding membership functions are determined for each indicator from the databases in order to formalise the indicators [29,34].

Therefore, mathematical models for assessment of the degree of anatomical lesion of the coronary arteries have the following form (1 ÷ 4):(1)$$\begin{array}{l} \mu^{d1}(X_{1},X_{2},X_{3},X_{4})=\mu^{HA}(X_{1})\cdot \mu^{A}(X_{2})\cdot \mu^{L}(X_{3})\cdot \mu^{A}(X_{4})\cdot \cup \\ \mu^{HA}(X_{1})\cdot \mu^{HA}(X_{2})\cdot \mu^{L}(X_{3})\cdot \mu^{A}(X_{4})\\ \cup \mu^{HA}(X_{1})\cdot \mu^{HA}(X_{2})\cdot \mu^{L}(X_{3})\cdot \mu^{HA}(X_{4}); \end{array}$$
(2)$$\begin{array}{l} \mu^{d2}(X_{1},X_{2},X_{3},X_{4})=\mu^{LA}(X_{1})\cdot \mu^{A}(X_{2})\cdot \mu^{L}(X_{3})\cdot \mu^{H}(X_{4})\cdot \\ \cup \mu^{A}(X_{1})\cdot \mu^{HA}(X_{2})\cdot \mu^{L}(X_{3})\cdot \mu^{LA}(X_{4})\cup \\ \mu^{LA}(X_{1})\cdot \mu^{HA}(X_{2})\cdot \mu^{L}(X_{3})\cdot \mu^{LA}(X_{4})\cup \mu^{A}(X_{1})\cdot \mu^{L}(X_{2})\cdot \mu^{L}(X_{3})\cdot \mu^{LA}(X_{4}); \end{array}$$
(3)$$\begin{array}{l} \mu^{d3}(X_{1},X_{2},X_{3},X_{4})=\mu^{HA}(X_{1})\cdot \mu^{LA}(X_{2})\cdot \mu^{HA}(X_{3})\cdot \mu^{H}(X_{4})\cdot \\ \cup \mu^{HA}(X_{1})\cdot \mu^{LA}(X_{2})\cdot \mu^{HA}(X_{3})\cdot \mu^{H}(X_{4})\cup \\ \mu^{L}(X_{1})\cdot \mu^{A}(X_{2})\cdot \mu^{HA}(X_{3})\cdot \mu^{H}(X_{4}); \end{array}$$
(4)$$\begin{array}{l} \mu^{d4}(X_{1},X_{2},X_{3},X_{4})=\mu^{L}(X_{1})\cdot \mu^{L}(X_{2})\cdot \mu^{A}(X_{3})\cdot \mu^{A}(X_{4})\cdot \\ \cup \mu^{LA}(X_{1})\cdot \mu^{L}(X_{2})\cdot \mu^{A}(X_{3})\cdot \mu^{A}(X_{4})\cup \\ \mu^{L}(X_{1})\cdot \mu^{LA}(X_{2})\cdot \mu^{LA}(X_{3})\cdot \mu^{LA}(X_{4})\cup \mu^{LA}(X_{1})\cdot \mu^{LA}(X_{2})\cdot \mu^{LA}(X_{3})\cdot \mu^{LA}(X_{4}); \end{array}$$

Each of the above-mentioned terms is a fuzzy set, specified by means of special membership functions and could be presented by a certain interval, that possesses its digital stages from 0 to 1. Absolute non-membership to the set is testified by 0, whereas absolute membership is testified by 1.

Appropriate membership functions for the formalisation of indices have been determined [33,44,45]. Therefore, logical equations for assessing the degree of anatomical lesion of the coronary arteries in patients with various forms of coronary artery disease will have the following form for the factors (X1 ÷ X4).

For factor X_1_
$$\begin{array}{l} {\tilde{\mu }}^{L}(X_{1})=\left\{ \begin{array}{l} 4.97-2.17x_{1},x_{1}\in [1.83;2.06)\\ 1.97-0.71x_{1},x_{1}\in [2.06;2.76] \end{array}\right.;\;{\tilde{\mu }}^{LA}(X_{1})=\left\{ \begin{array}{l} 2.17x_{1}-3.48,x_{1}\in [1.83;2.06)\\ 5.29-2.08x_{1},x_{1}\in [2.06;2.3)\\ 3.0-1.09x_{1},x_{1}\in [2.3;2.76] \end{array}\right.;\\ {\tilde{\mu }}^{A}(X_{1})=\left\{ \begin{array}{l} 2.13x_{1}-3.89,x_{1}\in [1.83;2.3)\\ 6-2.17x_{1},x_{1}\in [2.3;2.76] \end{array}\right.;\;{\tilde{\mu }}^{HA}(X_{1})=\left\{ \begin{array}{l} 1.06x_{1}-1.94,x_{1}\in [1.83;2.3)\\ 2.17x_{1}-4.5,x_{1}\in [2.3;2.53)\\ 6.5-2.17x_{1},x_{1}\in [2.53;2.76] \end{array}\right.;\\ {\tilde{\mu }}^{H}(X_{1})=\left\{ \begin{array}{l} 0.71x_{1}-1.31,x_{1}\in [1.83;2.53)\\ 2.17x_{1}-5,x_{1}\in [2.53;2.76] \end{array}\right.. \end{array}$$

For factor X_2_
$$\begin{array}{l} {\tilde{\mu }}^{L}(X_{2})=\left\{ \begin{array}{l} 3.72-1.56x_{2},x_{2}\in [1.74;2.06)\\ 1.6-0.53x_{2},x_{2}\in [2.06;3.0] \end{array}\right.;\;{\tilde{\mu }}^{LA}(X_{2})=\left\{ \begin{array}{l} 1.56x_{2}-2.22,x_{2}\in [1.74;2.06)\\ 4.3-1.61x_{2},x_{2}\in [2.06;2.37)\\ 2.38-0.79x_{2},x_{2}\in [2.37;3.0] \end{array}\right.;\\ {\tilde{\mu }}^{A}(X_{2})=\left\{ \begin{array}{l} 1.59x_{2}-2.76,x_{2}\in [1.74;2.3)\\ 4.76-1.59x_{2},x_{2}\in [2.3;3.0] \end{array}\right.;\;{\tilde{\mu }}^{HA}(X_{2})=\left\{ \begin{array}{l} 0.79x_{2}-1.38,x_{2}\in [1.74;2.37)\\ 1.56x_{2}-3.2,x_{2}\in [2.37;2.69)\\ 5.34-1.61x_{2},x_{2}\in [2.69;3.0] \end{array}\right.;\\ {\tilde{\mu }}^{H}(X_{2})=\left\{ \begin{array}{l} 0.53x_{2}-0.92,x_{2}\in [1.74;2.53)\\ 1.61x_{2}-3.84,x_{2}\in [2.53;3.0] \end{array}\right.. \end{array}$$

For factor X_3_
$$\begin{array}{l} {\tilde{\mu }}^{L}(X_{3})=\left\{ \begin{array}{l} 2.47-1.47x_{3},x_{3}\in [1.0;1.34)\\ 1.16-0.495x_{3},x_{3}\in [1.34;2.35] \end{array}\right.;\;{\tilde{\mu }}^{LA}(X_{3})=\left\{ \begin{array}{l} 1.47x_{3}-0.97,x_{3}\in [1.0;1.34)\\ 2.97-1.47x_{3},x_{3}\in [1.34;1.68)\\ 1.75-0.74x_{3},x_{3}\in [1.68;2.35] \end{array}\right.;\\ {\tilde{\mu }}^{A}(X_{3})=\left\{ \begin{array}{l} 1.47x_{3}-1.47,x_{3}\in [1.0;1.68)\\ 3.5-1.49x_{3},x_{3}\in [1.68;2.35] \end{array}\right.;\;{\tilde{\mu }}^{HA}(X_{3})=\left\{ \begin{array}{l} 0.73x_{3}-0.73,x_{3}\in [1.0;1.68)\\ 1.47x_{3}-1.97,x_{3}\in [1.68;2.02)\\ 4.06-1.52x_{3},x_{3}\in [2.02;2.35] \end{array}\right.;\\ {\tilde{\mu }}^{H}(X_{3})=\left\{ \begin{array}{l} 0.49x_{3}-0.49,x_{3}\in [1.0;2.02)\\ 1.52x_{3}-2.56,x_{3}\in [2.02;2.35] \end{array}\right.. \end{array}$$

For factor X_4_
$$\begin{array}{l} {\tilde{\mu }}^{L}(X_{4})=\left\{ \begin{array}{l} 2.59-0.769x_{4},x_{4}\in [2.07;2.72)\\ 1.197-0.26x_{4},x_{4}\in [2.72;4.67] \end{array}\right.;\;{\tilde{\mu }}^{LA}(X_{4})=\left\{ \begin{array}{l} 0.769x_{4}-1.09,x_{4}\in [2.07;2.72)\\ 3.09-0.769x_{4},x_{4}\in [2.72;3.37)\\ 1.796-0.38x_{4},x_{4}\in [3.37;4.67] \end{array}\right.;\\ {\tilde{\mu }}^{A}(X_{4})=\left\{ \begin{array}{l} 0.769x_{4}-1.59,x_{4}\in [2.07;3.37)\\ 3.59-0.769x_{4},x_{4}\in [3.37;4.67] \end{array}\right.;\;{\tilde{\mu }}^{HA}(X_{4})=\left\{ \begin{array}{l} 0.38x_{4}-0.796,x_{4}\in [2.07;3.37)\\ 0.769x_{4}-2.09,x_{4}\in [3.37;4.02)\\ 4.09-0.769x_{4},x_{4}\in [4.02;4.67] \end{array}\right.;\\ {\tilde{\mu }}^{H}(X_{4})=\left\{ \begin{array}{l} 0.25x_{4}-0.53,x_{4}\in [2.07;4.02)\\ 0.769x_{4}-2.59,x_{4}\in [4.02;4.67] \end{array}\right.. \end{array}$$

## 4. Discussion

For the construction of the equation, it is necessary to determine the membership functions *μ^j^*(*x_i_*) of all fuzzy terms j (H, HA, A, LA, L) for all factors *X_i_* (in the given case *j*-value of symmetry coefficient, *i*—interval of the study; $i=\bar{1,4}$). If we assume the high level to be the variant of the norm, then the construction of the equations is necessary to perform for five fuzzy terms (H, HA, A, LA, L). 

Each factor *X_i_* must correspond to its five membership functions. Certain actions should be taken to simplify modelling. Let $\underline{x_{i}}$ and $\bar{x_{i}}$—lower and upper boundaries of factor X_i_ changes range. We express the interval [$\underline{x_{i}}$, $\bar{x_{i}}$] on the interval U = [0, 4], where membership function ${\tilde{\mu }}^{j}(u)$, *u*
∈ U are set for fuzzy terms j = H, HA, A, LA, L). 

The graphic form of membership functions is shown in Figure 3. The choice of similar curves is stipulated by the fact that they are spline—linear approximations of expert membership function *μ^j^*(*x_i_*), obtained for factors *x*_1_ ÷ *x*_4_ by means of the method of paired comparisons [33].

The transition from the function ${\tilde{\mu }}^{j}(u)$ to required functions *μ^j^*(*x_i_*) is performed in the following way [33,40,45,46,47,48].
$$u_{i}=4\frac{x_{n}-\underline{x_{n}}}{\bar{x_{n}}-\underline{x_{n}}},\;{\tilde{\mu }}^{j}(u_{n})=\mu^{j}(x_{n}).$$

Decision making regarding the degree of the disease severity could be performed according to the following algorithm [41,49,50,51]. 

Step 1: Fix the value of factors for the given patient *x_n_* ($n=\bar{1,4}$);

Step 2: Applying the Formulas (1)–(4), we determine the values of membership functions *μ^j^*(*x_n_*) at fixed values of X_n_ factors;

Step 3: Using logic equations, we calculate membership functions $\mu^{d_{i}n}(x_{1},x_{2},\ldots ,x_{n})$ for all the stages of the disease severity d_n_, $n=\bar{1,4}$. Operations AND(**·**) and OR (V) over membership functions *μ*(*a*) and *μ*(*ϐ*) are replaced by the operations min and max.
*μ*(*a*) *• μ*(*ϐ*) *= min*[*μ*(*a*), *μ*(*ϐ*)]; *μ*(*a*)*Vμ*(*ϐ*) = *max*[*μ*(*a*), *μ*(*ϐ*)];

Step 4: The decision do is made, for which (5)
(5)$$\mu^{d_{0}}(x_{1},x_{2},\ldots ,x_{n})=\max[\mu^{d_{n}}(x_{1},x_{2},\ldots ,x_{n})]$$

This decision will correspond to the required range that indicates the degree of the disease severity. 

In the process of biomedical research, the problem of adjusting the neurofuzzy network appears. For the adjustment of this network parameters the recurrent relations, suggested by Professor O.P. Rotshtein, are used [33,45,46]. The essence of the adjustment is the selection of such parameters of membership functions ($b_{i}^{jp}(t)$, $c_{i}^{jp}(t)$) and weights of fuzzy rules ($w_{jp}(t)$) that provide minimum divergence between models and diagnostic results (see Figure 4).
$$\sum_{i=1}^{M}(F_{y}({\overset{⌢}{x}}_{1}^{l},{\overset{⌢}{x}}_{2}^{l},\ldots ,{\overset{⌢}{x}}_{12}^{l},W_{i})-{\overset{⌢}{y}}_{l}{)}^{2}=\underset{W_{i}}{\min},$$
where $\langle {\overset{⌢}{X}}_{l},{\overset{⌢}{y}}_{l}\rangle ,l=\bar{1,M}$—data of experimental research; b—maximum coordinate; C—parameter of compression and extension [52,53,54].

### Practical Implementation of the Medical Expert System for Assessing the Degree of Anatomical Lesion of the Coronary Arteries in Patients with Various Forms of Coronary Artery Disease

The main ideology of the implementation of the medical expert system for assessing the degree of anatomical lesion of the coronary arteries in patients with various forms of coronary artery disease based on the formation of fuzzy logic blocks is shown in Figure 5.

As a result of the implementation of these blocks, a software shell was developed, after starting the program, the user is invited to enter the values of the upper and lower scale of values that are in the database for a certain pathology, which are the main factors for the determination of the degree of anatomical lesion of coronary arteries [34,47,48].

The result of the implementation of these blocks was a software shell that works as follows (see Figure 6 and Figure 7).

After starting the program, the user is asked to enter the values of the upper and lower scale of values that are in the database on a certain basis based on the face size indicators, in our case we enter the values that are the main ones when determining [55,56,57].To continue working with the program, after filling in all the fields, press “Save”, to restore previous data that were entered before, the user needs to click “Retire”.

The validity of the support of the correct decision making of the medical expert system on the basis of fuzzy sets for the assessment of the degree of anatomical lesion of the coronary artery in patients with various forms of coronary disease with the assessment of experts was 95%, which shows the high efficiency of the decision making.

## 5. Conclusions

The method of using fuzzy sets for the implementation of an information expert system for solving the problems of medical diagnostics, in particular, when assessing the degree of anatomical lesion of the coronary arteries in patients with various forms of coronary artery disease, has been developed.

The paper analyses the main areas of application of mathematical methods in medical diagnostics and formulates the principles of diagnostics based on fuzzy logic.

Main scientific results: mathematical models and algorithms were developed that formalise the process of making diagnostic decisions based on fuzzy logic with quantitative and qualitative parameters of the patient’s condition; mathematical models of membership functions that formalise the representation of quantitative and qualitative parameters of the patient’s condition in the form of fuzzy sets, which are used in models and algorithms for assessing the degree of anatomical lesion of the coronary arteries in patients with various forms of coronary artery disease were developed.

The developed models and algorithms of medical diagnostics are based on the ideas and principles of artificial intelligence and knowledge engineering, the theory of experiment planning, theory of fuzzy sets and linguistic variables. The expert system is tested on real data.

The practical value of the work lies in the possibility of using the automated expert system to solve the problems of medical diagnosis based on fuzzy logic when assessing the degree of anatomical lesion of the coronary arteries in patients with various forms of coronary artery disease.

## Figures and Tables

**Figure 1 ijerph-20-00979-f001:**
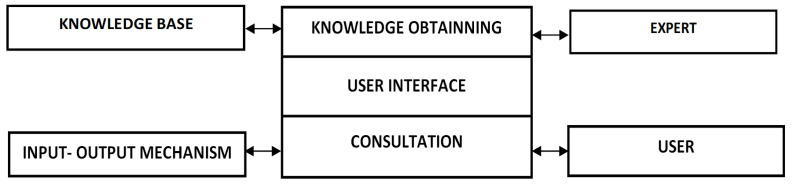
Basic structure of MES [40].

**Figure 2 ijerph-20-00979-f002:**
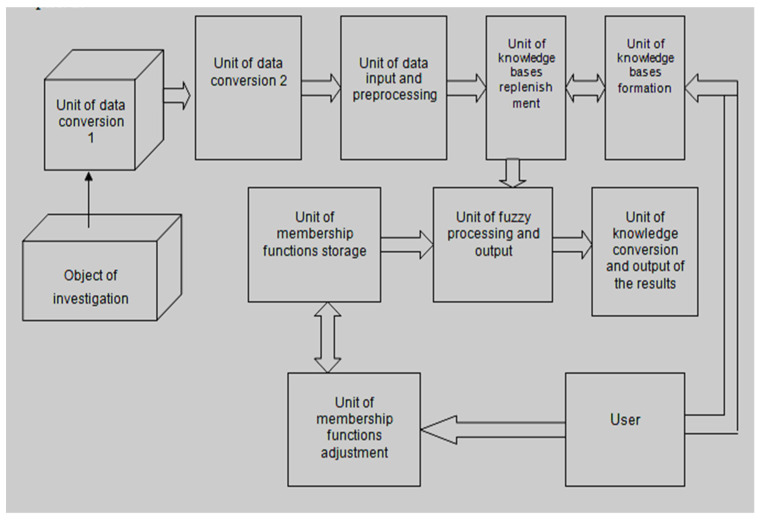
Architecture of the medical expert system.

**Figure 3 ijerph-20-00979-f003:**
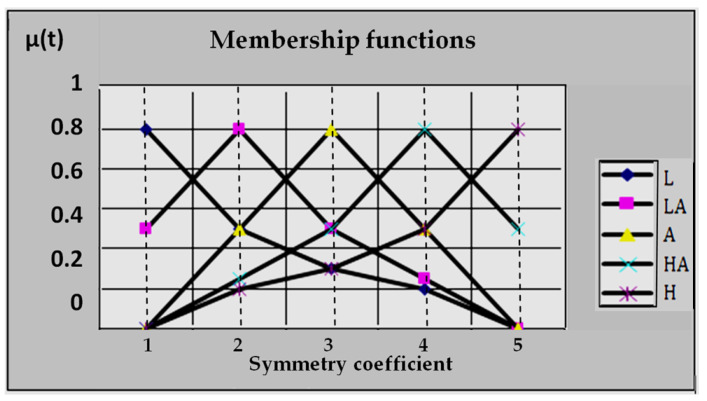
Membership functions of fuzzy terms.

**Figure 4 ijerph-20-00979-f004:**
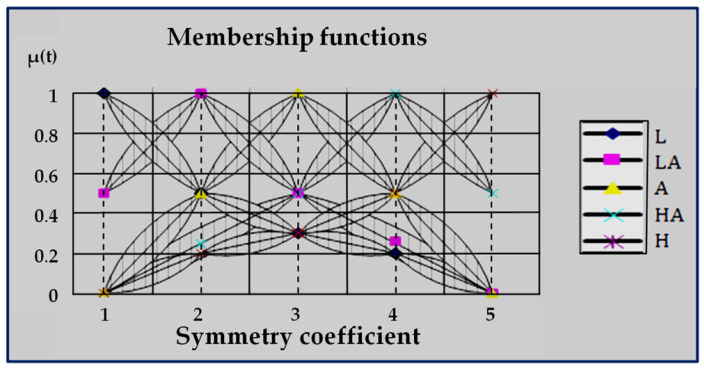
Membership functions of fuzzy terms after the tuning procedure [33].

**Figure 5 ijerph-20-00979-f005:**
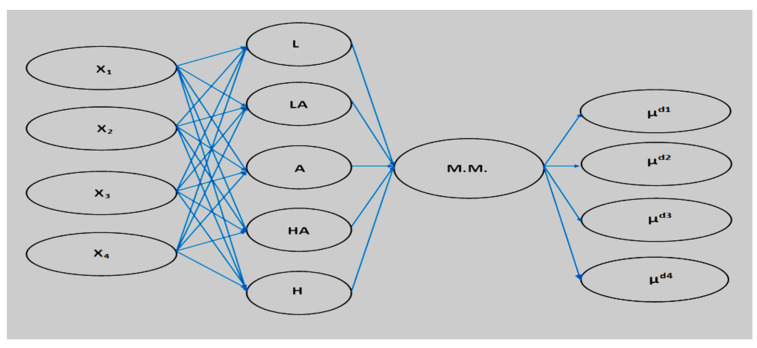
Medical expert system for determining the degree of anatomical lesion of the coronary arteries.

**Figure 6 ijerph-20-00979-f006:**
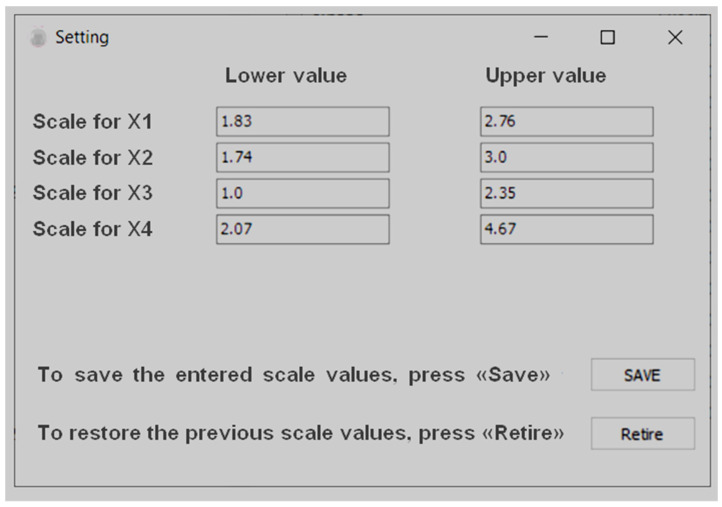
Function of entering minimum and maximum values for each factor.

**Figure 7 ijerph-20-00979-f007:**
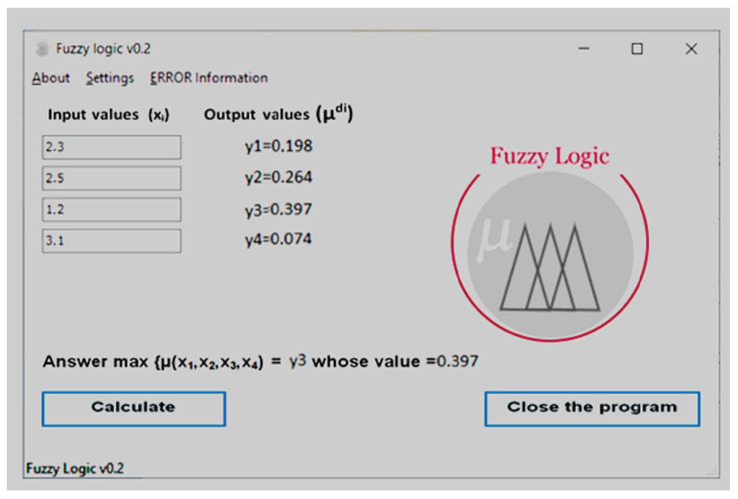
An example of a dialogue window of a program system for determining the degree of anatomical lesion of the coronary arteries.

**Table 1 ijerph-20-00979-t001:** Statistics of anatomical lesions of the coronary arteries in patients with various forms of CAD.

Peculiarities of Anatomic Lesions of the Coronary Arteries	Clinical Forms of CAD			
	1. NSTEMI	2. UAP	3. STEMI	4. StAP
1	2	3	4	5
Number of patients	**90**	**25**	**25**	**25**
Presence of a/p in the basin of the LCA trunk	9 (10.0%)	1 (4.0%)	4 (16.0%)	2 (8.0%)
Presence of HSS in the trunk of the LCA	-	-	-	-
Presence of a/p on the territory of DG or LAD LCA	62 (68.9%)	8 (32.0%)	21 (84.0%)	8 (32.0%)
*P* according to the *χ*^2^	p1–2 = 0.001; p1–4 = 0.001; p2–3 < 0.0001; p3–4 < 0.0001			
Presence of HSS in the basin of DG or LAD LCA	56 (62.2%)	8 (32.0%)	15 (60.0%)	6 (24.0%)
*P* according to the *χ*^2^	p1–2 = 0.007; p1–4 = 0.001; p2–3 = 0.047; p3–4 = 0.01			
Severity of stenosis in points	**2.57 ± 0.07**	**2.15 ± 0.13**	**2.64 ± 0.12**	**2.00 ± 0.17**
*One-way ANOVA and LSD test*	p1–2 = 0.006; p1–4 = 0.003; p2–3 = 0.008; p3–4 = 0.003			
Presence of HSS in the basin of LCx	32 (35.6%)	2 (8.0%)	12 (48.0%)	6 (24.0%)
*P* according to the *χ*^2^	p1–2 = 0.0006; p2–3 = 0.002			
Presence of HSS in the basin of LCx	32 (35.6%)	2 (8.0%)	7 (28.0%)	5 (20.0%)
*P* according to the *χ*^2^	p1–2 = 0.008			
Severity of stenosis in points	**2.42 ± 0.18**	**2.50 ± 0.50**	**2.08 ± 0.23**	**2.00 ± 0.26**
Presence of a/p in the basin of the RCA trunk	29 (32.2%)	2 (8.0%)	18 (72.0%)	6 (24.0%)
*P according* *to* *the criterion χ* ^2^	p1–2 = 0.02; p1–3 < 0.0001; p2–3 < 0.0001; p3–4 = 0.001			
Presence of HSS in the trunk of the RCA	22 (24.4%)	0 (0)	16 (64.0%)	5 (20.0%)
*P* according to the *χ*^2^	p1–2 = 0.006; p1–3 < 0.0001; p2–3 < 0.0001; p2–4 = 0.02; p3–4 = 0.002			
Severity of stenosis in points	**2.23 ± 0.12**	**1.00 ± 0**	**2.11 ± 0.14**	**2.00 ± 0.26**
*One-way ANOVA and LSD test*	p1–2 < 0.0001; p2–3 < 0.0001; p2–4 = 0.0003			
Absence of HSS CA	13 (14.4%)	16 (64.0%)	0 (0)	15 (60.0%)
*P* according to the *χ*^2^	p1–2 < 0.0001; p1–3 = 0.04; p1–4 < 0.0001; p2–3 < 0.0001; p3–4 < 0.0001			
HSS single-vessel lesion	55 (61.1%)	8 (32.0%)	14 (56.0%)	6 (24.0%)
*P* according to the *χ*^2^	p1–2 = 0.01; p1–4 = 0.001 p3–4 = 0.02			
HSS two-vessel lesion	16 (17.8%)	1 (4.0%)	9 (36.0%)	2 (8.0%)
*P* according to the *χ*^2^	p1–3 = 0.05; p2–3 = 0.005 p3–4 = 0.02			
HSS three-vessel lesion	9 (10.0%)	0 (0)	2 (8.0%)	2 (8.0%)
Severity of CA lesion, total score	**3.66 ± 0.20**	**2.50 ± 0.43**	**4.32 ± 0.35**	**3.64 ± 0.43**
*One-way ANOVA and LSD test*	p1–2 = 0.02; p2–3 = 0.002			

**Note.** Here and in the following tables: a/p—atherosclerotic plaques, NSTEMI—myocardial infarction without ST segment elevation, UAP—unstable angina pectoris, STEMI—myocardial infarction with ST segment elevation, StAP—stable tension angina pectoris, CA—coronary arteries, HSS—hemodynamically significant stenosis (>50%), LCA—left coronary artery, DG—diagonal and LAD—left anterior descending artery, RCA—right coronary artery, LCx—left circumflex artery.

**Table 2 ijerph-20-00979-t002:** Main clinical characteristics of NSTEMI patients on the whole by group and depending on gender.

Clinical Characteristics (n = 200)	Men (n = 142)	Women (n = 58)	*p*
HD, number (%) **n = 171 (85.5%)**	120 (84.5%)	51 (87.9%)	0.53
Duration of HD, years **[7,8,9,10,11,12,13,14,15,16,17,18,19,20,21,22,23,42,43] 15.5 ± 0.41**	15.3 ± 0.50	15.8 ± 0.71	0.63
Tension angina pectoris I–III FC to MI, number (%) **n = 86 (43.0%)**	54 (38.0%)	32 (55.2%)	**0.03**
Duration of angina pectoris, years **[1,2,3,4,5,6,7,8,9,10,11,12,13,14,15] 7.0 ± 0.44**	6.8 ± 0.60	7.3 ± 0.63	0.67
Permanent form of AF, number (%) **n = 23 (11.5%)**	10 (7.0%)	13 (22.4%)	**0.002**
Duration of constant AF, years **[1,2,3,4,5,6,7] 4.4 ± 0.39**	4.4 ± 0.66	4.3 ± 0.50	0.98
DM type II, number (%) **n = 25 (12.5%)**	16 (11.3%)	9 (15.5%)	0.40
Smoking, number (%) **n = 84 (42.0%)**	74 (52.1%)	10 (17.2%)	**<0.0001**
Active smoking history, years **[14,15,16,17,18,19,20,21,22,23,24,25,26,27,28,29,30,31,32,33,34,35,36,37,38,42,43] 29.5 ± 0.84**	29.6 ± 0.75	25.0 ± 1.01	**0.002**
Alimentary obesity, number (%) **n = 73 (36.5%)**	51 (35.9%)	22 (37.9%)	0.78
I degree (BMI—30–35 kg/m^2^) **n = 51 (25.5%)**	37 (26.1%)	14 (24.1%)	0.77
II degree (BMI—35–40 kg/m^2^) **n = 18 (9.0%)**	12 (8.5%)	6 (10.3%)	0.67
III degree (BMI > 40 kg/m^2^) **n = 4 (2.0%)**	2 (1.4%)	2 (3.4%)	0.34
BMI, kg/m^2^ **[19.3–47.6] 28.6 ± 0.36**	28.4 ± 0.41	29.0 ± 0.74	0.42

**Notes:** AH—hypertension disease, MI—myocardial infarction, AF—atrial fibrillation, DM—type II diabetes, BMI—mass index; the minimum and maximum values of the indicator are given in square brackets—[min–max]; the reliability of the difference in per cent between men and women was calculated according to the χ^2^ criterion and between the average values of indicators—according to the T-test for independent samples by groups.

**Table 3 ijerph-20-00979-t003:** Features of the anatomical lesions of the coronary arteries in patients with various forms of CAD.

Clinical Forms of CAD	Features of the Anatomical Lesion Coronary Arteries			
	Presence of a/p on the Territory of DG or LAD LCA X1	Presence of HSS on the Territory of DG or LAD LCA X2	(Presence of HSS in the Trunk of the RCA X3	Absence of HSS CA X4
**NSTEMI myocardial infarction without ST segment elevation** **µ^I^(x_1_x_2_x_3_x_4_)**	2.57 ± 0.07	2.42 ± 0.18	2.23 ± 0.12	3.66 ± 0.20
**UAP—unstable angina pectoris** **µ^II^(x_1_x_2_x_3_x_4_)**	2.15 ± 0.13	2.50 ± 0.50	1.00 ± 0	2.50 ± 0.43
**STEMI—myocardial infarction with ST segment elevation** **µ^III^(x_1_x_2_x_3_x_4_)**	2.64 ± 0.12	2.08 ± 0.23	2.11 ± 0.14	4.32 ± 0.35
**StAP—stable angina pectoris** **µ^IV^(x_1_x_2_x_3_x_4_)**	2.00 ± 0.17	2.00 ± 0.26	2.00 ± 0.26	3.64 ± 0.43
	**min/max** **1.83 ÷ 2.76**	**min/max** **1.74 ÷ 3.0**	**min/max** **1.0 ÷ 2.35**	**min/max** **2.07 ÷ 4.67**

**Table 4 ijerph-20-00979-t004:** Database regarding ratios.

Clinical Forms of CAD	Features of the Anatomical Lesion of the Coronary Arteries			
	X1	X2	X3	X4
**µ^I^(x_1_x_2_x_3_x_4_)**	HA	A	L	A
	HA	HA	L	A
	HA	HA	L	HA
**µ^II^(x_1_x_2_x_3_x_4_)**	LA	A	L	H
	A	HA	L	LA
	LA	HA	L	LA
	A	L	L	LA
**µ^III^(x_1_x_2_x_3_x_4_)**	HA	LA	HA	HA
	HA	LA	HA	H
	L	A	HA	H
**µ^IV^(x_1_x_2_x_3_x_4_)**	L	L	A	A
	LA	L	A	A
	L	LA	LA	LA
	LA	LA	LA	LA

## Data Availability

The data are not publicly available due to privacy and ethical restrictions. The data presented in this study may be available conditionally from the corresponding author.

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
