# Peer review of "Medical Fuzzy-Expert System for Assessment of the Degree of Anatomical Lesion of Coronary Arteries"

_ijerph, 2023, doi:10.3390/ijerph20020979_

Round 1

Reviewer 1 Report

The study concerns an expert system evaluating the degree of coronary artery lesion in patients with coronary artery disease. The aim of the reaserch has been achieved. After additional verifiction, the propoed model can be used as an automated expert system in medical daignostic.

Author Response

Dear Reviewer,

The authors thank the reviewers for their valuable comments and suggestions that improved the quality and structure of the paper.

Comments and Suggestions for Authors

The study concerns an expert system evaluating the degree of coronary artery lesion in patients with coronary artery disease. The aim of the reaserch has been achieved. After additional verifiction, the propoed model can be used as an automated expert system in medical daignostic.

Thank you very much for your favorable review.

Sincerely

Anna Lewandowska

Reviewer 2 Report

This research seeks to build a fuzzy-sets-based medical expert system for CABG patients. This study uses a method of fuzzy sets to generate an information expert system for medical diagnostics, especially coronary artery lesion analysis, which has been developed. The study has successfully addressed mathematical and medical applications and fuzzy logic diagnostic ideas. Through research and comparison of expert and medical expert system findings, the dependability of supporting proper decision-making of the medical expert system based on fuzzy sets was 95%, exhibiting good decision-making efficiency. The practical usefulness of the work lies in the possibility of using an automated expert system to assess coronary artery lesions in individuals with coronary artery disease using fuzzy logic. The proposed idea needs inter-rater consistency and dependability.

 Here are the reviewer's general comments and questions:

(+ ) The authors have effectively used logical connections to create a clear flow between ideas, allowing the reader to make sense of the content.

(+ ) The authors have also used appropriate language for the subject matter, avoiding ambiguity and confusion.

 (-) The authors should use transitions to connect topics and ensure a smooth reading experience.

Overall, this scientific article writing can improve and communicate its ideas more effectively.

 Line 106-113; Line 149-156: The first letter of each line should be capitalized.

Line 226-229: A scientific article must be well structured. Consider eliminating the numbers 1, 2, and 3 in lines 227, 228, and 229. It doesn't make sense to use orders 1, 2, and 3 since I can't see any subtitles here.

Line 186-220 The number and order make the reviewer confused. Please consider the coherence of a paper's structure based on titles and subtitles.

Line 445-466: A scientific article must be well structured. Consider eliminating the numbers 1, 2, 3, 4, and 5 from lines 445 to 466. It doesn't make sense to use the order 1, 2, 3, 4, and 5 since I can't see any subtitles here. Every conclusion can be put into different paragraphs.

Figure 1, Figure 2, Figure 3, Figure 4, Figure 5, Figure 6, Figure 7: shallow resolution. The author should increase the resolution of these pictures.

Why did you choose the proposed methods in your research? Have you compared its accuracy with that of other approaches?

Which data source you used to support this research? How do you evaluate the reliability of this data source?

Author Response

Dear Reviewer,

We would like to thank the reviewers for their comments. After analysing all the comments, we made the following changes:

Comments and Suggestions for Authors

This research seeks to build a fuzzy-sets-based medical expert system for CABG patients. This study uses a method of fuzzy sets to generate an information expert system for medical diagnostics, especially coronary artery lesion analysis, which has been developed. The study has successfully addressed mathematical and medical applications and fuzzy logic diagnostic ideas. Through research and comparison of expert and medical expert system findings, the dependability of supporting proper decision-making of the medical expert system based on fuzzy sets was 95%, exhibiting good decision-making efficiency. The practical usefulness of the work lies in the possibility of using an automated expert system to assess coronary artery lesions in individuals with coronary artery disease using fuzzy logic. The proposed idea needs inter-rater consistency and dependability.

Here are the reviewer's general comments and questions:

(+ ) The authors have effectively used logical connections to create a clear flow between ideas, allowing the reader to make sense of the content.

(+ ) The authors have also used appropriate language for the subject matter, avoiding ambiguity and confusion.

(-) The authors should use transitions to connect topics and ensure a smooth reading experience.

We improved transitions with counters.

Overall, this scientific article writing can improve and communicate its ideas more effectively.

 Line 106-113; Line 149-156: The first letter of each line should be capitalized.

Capitalized spelling corrected

Line 226-229: A scientific article must be well structured. Consider eliminating the numbers 1, 2, and 3 in lines 227, 228, and 229. It doesn't make sense to use orders 1, 2, and 3 since I can't see any subtitles here.

Line 186-220 The number and order make the reviewer confused. Please consider the coherence of a paper's structure based on titles and subtitles.

Line 445-466: A scientific article must be well structured. Consider eliminating the numbers 1, 2, 3, 4, and 5 from lines 445 to 466. It doesn't make sense to use the order 1, 2, 3, 4, and 5 since I can't see any subtitles here. Every conclusion can be put into different paragraphs.

Figure 1, Figure 2, Figure 3, Figure 4, Figure 5, Figure 6, Figure 7: shallow resolution. The author should increase the resolution of these pictures.

The authors corrected comments on the design and structuring of the paper, improved the quality of the figures.

Why did you choose the proposed methods in your research? Have you compared its accuracy with that of other approaches?

Which data source you used to support this research? How do you evaluate the reliability of this data source?

The authors additionally justified the choice of the proposed research method and cited the advantages of the proposed method for assessment of the degree of anatomical lesion of the coronary arteries.

They also cited scientific sources to support this research and showed the thoroughness and reliability of the sources they relied on.

I hope that the changes made are satisfactory and this will allow publication. I am asking you to take into account the positive comments of the reviewers that this is an interesting study and a good study.

Sincerely

Anna Lewandowska

Reviewer 3 Report

In this paper, the authors introduced a medical expert system based on fuzzy sets for assessing the degree of coronary artery lesion in patients with coronary artery disease. Methods: The method of using fuzzy sets for the implementation of an information expert system for solving the problems of medical diagnostics, in particular, when assessing the degree of anatomical lesion of the coronary arteries in patients with various forms of coronary artery disease, has been developed. Results: The paper analyzes the main areas of application of mathematical methods in medical diagnostics, formulates the principles of diagnostics, based on fuzzy logic. The developed models and algorithms of medical diagnostics are based on the ideas and principles of artificial intelligence and knowledge engineering, theory of experiment planning, theory of fuzzy sets and linguistic variables. The expert system is tested on real data

The results are novel, mathematically true, and interesting. The paper is suitable for publication in IJERPH. However, the following minor revisions are needed before the final version:

1.     This manuscript needs further improvement in sentence expression because of a few grammar mistakes.

2.     The motivation and contribution of the proposed work are not given in the manuscript. Authors should include a subsection that clearly specifies the proposed work's motivation and objective. Novelty in the proposed work should be explicitly highlighted. It isn't easy to see the Novelty of this work.

3.     Related work and existing literature need to be better presented. It is suitable to describe existing studies, focus areas and problems identified from existing literature.

4.     Please give more references about this topic and improve the importance of this paper by comparing existing literature.

5.     Please cite some related references from the journal to better suitability of the paper in this journal, like

Rześny-Cieplińska, J., Szmelter-Jarosz, A. & Moslem, S. (2021). Priority-Based Stakeholders Analysis in the View of Sustainable City Logistics: Evidence for Tricity, Poland. Sustainable Cities and Society, 67, 102751

Moslem, S., Farooq, D., Ghorbanzadeh, O., & Blaschke, T. (2020). Application of the AHP-BWM Model for Evaluating Driver Behavior Factors Related to Road Safety: A Case Study for Budapest. Symmetry, 12 (2), 243.

Duleba, S., Moslem, S. (2021). User Satisfaction Survey on Public Transport by a New PAHP Based Model. Applied Sciences, , 11(21), 10256. 

https://link.springer.com/book/10.1007/978-981-19-4929-6.

6.      The abstract needs to be clarified. 

Author Response

Dear Reviewer,

We would like to thank the reviewers for their comments. After analysing all the comments, we made the following changes:

Comments and Suggestions for Authors

In this paper, the authors introduced a medical expert system based on fuzzy sets for assessing the degree of coronary artery lesion in patients with coronary artery disease. Methods: The method of using fuzzy sets for the implementation of an information expert system for solving the problems of medical diagnostics, in particular, when assessing the degree of anatomical lesion of the coronary arteries in patients with various forms of coronary artery disease, has been developed. Results: The paper analyzes the main areas of application of mathematical methods in medical diagnostics, formulates the principles of diagnostics, based on fuzzy logic. The developed models and algorithms of medical diagnostics are based on the ideas and principles of artificial intelligence and knowledge engineering, theory of experiment planning, theory of fuzzy sets and linguistic variables. The expert system is tested on real data

The results are novel, mathematically true, and interesting. The paper is suitable for publication in IJERPH. However, the following minor revisions are needed before the final version:

  1. This manuscript needs further improvement in sentence expression because of a few grammar mistakes.

Correct English

  1. The motivation and contribution of the proposed work are not given in the manuscript. Authors should include a subsection that clearly specifies the proposed work's motivation and objective. Novelty in the proposed work should be explicitly highlighted. It isn't easy to see the Novelty of this work.

The authors supplemented the article with a statement of the problem, provided motivation and showed the contribution of the work to the theory of the development of fuzzy sets in the context of creating an expert fuzzy medical system for assessment of the degree of anatomical lesion of the coronary arteries.

The authors defined the scientific Novelty and the aim of the work in the paper.

  1. Related work and existing literature need to be better presented. It is suitable to describe existing studies, focus areas and problems identified from existing literature.
  2. Please give more references about this topic and improve the importance of this paper by comparing existing literature.
  3. Please cite some related references from the journal to better suitability of the paper in this journal, like

Rześny-Cieplińska, J., Szmelter-Jarosz, A. & Moslem, S. (2021). Priority-Based Stakeholders Analysis in the View of Sustainable City Logistics: Evidence for Tricity, Poland. Sustainable Cities and Society, 67, 102751

Moslem, S., Farooq, D., Ghorbanzadeh, O., & Blaschke, T. (2020). Application of the AHP-BWM Model for Evaluating Driver Behavior Factors Related to Road Safety: A Case Study for Budapest. Symmetry, 12 (2), 243.

Duleba, S., Moslem, S. (2021). User Satisfaction Survey on Public Transport by a New PAHP Based Model. Applied Sciences, , 11(21), 10256. 

https://link.springer.com/book/10.1007/978-981-19-4929-6.

The authors have added more links, in particular, to papers from this journal, to research topics, and linked literature on the development of this scientific direction.

  1.  The abstract needs to be clarified. 

The authors corrected the abstract of the paper.

I hope that the changes made are satisfactory and this will allow publication. I am asking you to take into account the positive comments of the reviewers that this is an interesting study and a good study.

Sincerely

Anna Lewandowska
